# The Putative Cytochrome *b*5 Domain-Containing Protein CaDap1 Homologue Is Involved in Antifungal Drug Tolerance, Cell Wall Chitin Maintenance, and Virulence in *Candida albicans*

**DOI:** 10.3390/jof10050316

**Published:** 2024-04-26

**Authors:** Dayong Xu, Manman Wang, Xing Zhang, Hongchen Mao, Haitao Xu, Biao Zhang, Xin Zeng, Feng Li

**Affiliations:** College of Life Sciences, Huaibei Normal University, Huaibei 235000, China; 18956121307@163.com (M.W.); xzhang1995@126.com (X.Z.); m17805617867@163.com (H.M.); hsdhaitao@163.com (H.X.); zhangbiao@chnu.edu.cn (B.Z.); xzenghsd@163.com (X.Z.)

**Keywords:** *Candida albicans*, *CaDAP1*, drug susceptibility, chitin content, virulence

## Abstract

*Candida albicans* (*Ca*), a prominent opportunistic fungal pathogen in humans, has garnered considerable attention due to its infectious properties. Herein, we have identified and characterized *CaCDAP1* (*Ca orf19.1034*), a homolog of *ScDAP1* found in *Saccharomyces cerevisiae*. *CaCDAP1* encodes a 183-amino acid protein with a conserved cytochrome *b*5-like heme-binding domain. The deletion of *CaDAP1* renders *Ca* cells susceptible to caspofungin and terbinafine. *CaDAP1* deletion confers resistance to Congo Red and Calcofluor White, and sensitivity to sodium dodecyl sulfate. The deletion of *CaDAP1* results in a 50% reduction in chitin content within the cell wall, the downregulation of phosphorylation levels in CaMkc1, and the upregulation of phosphorylation levels in CaCek1. Notably, *CaDAP1* deletion results in the abnormal hyphal development of *Ca* cells and diminishes virulence in a mouse systemic infection model. Thus, *CaDAP1* emerges as a critical regulator governing cellular responses to antifungal drugs, the synthesis of cell wall chitin, and virulence in *Ca*.

## 1. Introduction

*Candida albicans* (*Ca*) stands out as the most prevalent opportunistic fungal pathogen in humans, existing as an innocuous commensal in approximately 70% of individuals [1,2]. However, in immunocompromised patients, it can transition to causing both superficial mucosal and potentially life-threatening systemic infections [3,4,5]. The pathogenicity of *Ca* is multifaceted, involving various factors and activities such as attachment to and penetration into host cells, the release of hydrolases, transition from yeast to hyphae, perception of contact, responsiveness to physical stimuli, alteration in phenotype, development of biofilms, adjustment to environmental acidity, flexibility in metabolism, efficient nutrient acquisition mechanisms, and reliable stress response systems [5,6].

The cell wall of *Ca* is composed of chitin, β-glucan, mannan, and cell wall proteins [7]. Chitin and β-glucan are located in the interior of the cell wall, and form the core skeleton structure. The outermost portion of the cell wall is mainly composed of mannan, which masks the internal β-glucan and reduces the recognition of *Ca* by the host immune system. The cell wall of *Ca* plays a key role in maintaining cell integrity, morphogenesis, response to changes in environmental conditions, interaction with the host, and pathogenesis [7,8]. Moreover, the cell wall of *Ca* is a prime target for antifungal drugs such as the echinocandins [9]. Despite its tough cell wall, *Ca* can flexibly change the relative levels of chitin, β-glucan, and mannan in response to environmental change [10,11]. Thus, this potential for cell wall remodeling is critical for maintaining *Ca* cell wall integrity (CWI) and is regulated by multiple signaling pathways, including the Mkc1, Hog1, and Cek1 mitogen-activated protein (MAP) kinase cascade [12]. Chitin is absent in humans and other vertebrates. Therefore, chitin synthesis is an excellent potential target for the development of antifungal drugs [13]. *Ca* regulates the expression of chitin synthase and the content of chitin in the cell wall, through HOG, PKC, and the Ca^2+^/calcineurin signaling pathways in response to environmental stress [13]. 

In the context of *Saccharomyces cerevisiae*, the heme-binding damage-resistance protein 1 (DAP1) is associated with cytochrome *b*5, and plays significant roles in iron metabolism and ergosterol biosynthesis [14,15,16,17,18]. In *Schizosaccharomyces pombe*, DAP1 positively regulates P450 enzymes and is essential for sterol biosynthesis [19]. In *Aspergillus fumigatus*, DapA influences the susceptibility to azoles by maintaining the stability of cytochrome P450 enzymes responsible for ergosterol synthesis [20]. Recognizing the significance of the heme-binding damage-resistance protein 1, we explored the *Ca* genome for genes homologous with *S. cerevisiae* ScDAP1. Our search revealed that the identity between *Ca orf19.1034* (*CaDAP1*) and *S. cerevisiae ScDAP1* (*YPL170W*) was 40%.

This study provides experimental evidence affirming the requirement of *CaDAP1* (*orf19.1034*) for antifungal drug tolerance and chitin synthesis in the cell wall of *Ca*. Additionally, we demonstrate that the *Cadap1/Cadap1* null mutant exhibits defective hyphal development and diminished virulence in a mouse systemic infection model. Our findings suggest that CaDap1 plays crucial roles in the morphogenesis and pathogenesis of *Ca*.

## 2. Materials and Methods

### 2.1. Strains, Media, Plasmids, and Primers

The *Ca* strains utilized in this investigation, including strains, plasmids and primers, are documented in Table 1 and Table 2. Strains were cultivated at 30 °C in either YPD medium or SD medium [21,22,23]. 

### 2.2. Construction of the Cadap1/Cadap1 Mutant and the Re-Integrant Strain (RS)

For the deletion of the 1st *CaDAP1* allele, a *SAT1* flipper cassette harboring *CaSAT1* was used for selection [25]. The *SAT1* flipper cassette, derived from pSFS2 using the polymerase chain reaction (PCR) with primer pairs DAP1 − NAT − UP and DAP1 − NAT − DOWN, was introduced into the RM1000 strain. The resultant *CaDAP1/Cadap1::SAT1-FLIP Nat^R^* transformant was chosen, and its genotypes were verified using PCR.

For the disruption of the second *CaDAP1* allele, the HIS1 cassette was amplified from pGEM − HIS1 using primer pairs DAP1 − HIS − UP and DAP1 − HIS − DOWN, followed by transformation into the *CaDAP1/Cadap1::SAT1* − *FLIP* strain. The resulting *Cadap1::HIS1/Cadap1::SAT1* − *FLIP Nat^R^* and *HIS^+^* transformants were chosen, and their genotypes validated using PCR. The *Cadap1::HIS1/Cadap1::SAT1 − FLIP Nat^R^* strain was grown on YPD medium supplemented with nourseothricin (25 μg/mL) to excise the Nat^R^ genetic biomarker, yielding the *Cadap1::HIS1/Cadap1::FRT* strain, whose genotype was verified by PCR using primers DAP1 − ORF − UP/DAP1 − ORF − DOWN and DAP1 − UP − CHECK/DAP1 − DOWN − CHECK (Figure 1).

To obtain a control, in this case a re-integrant strain (RS) for control purposes, the *CaDAP1* open reading frame, along with 800 and 281 bp of its promoter and terminator, respectively, were amplified by PCR using SC5314 genomic DNA as a template, with primers DAP1−UP−CLO and DAP1−DOWN−CLO. The 1.36 kb products obtained were cloned into the *Sac*I and *Not*I sites of CIp10. The resultant plasmids were digested with *Stu*I and transformed into the *Cadap1/Cadap1* mutant strain. Strain RM1000, transformed with *Stu*I−digested CIp10, served as a positive control, with expression of *URA3* from the *RPS1* locus (Table 1).

### 2.3. Growth Phenotypic Analysis and Hyphal Formation Examination

Plate assays were conducted following established protocols [23]. In brief, cells were cultured in liquid medium at 30 °C overnight, serially diluted to 5 × 10^7^, 5 × 10^6^, 5 × 10^5^, 5 × 10^4^, and 5 × 10^3^ cells/mL, and 2.5 µL samples grown on plates supplemented with terbinafine, caspofungin, fluconazole, SDS, CR, or CFW. Phenotypic observations were recorded after incubation at 30 °C for 48–72 h. Hyphal formation analysis was performed by culturing appropriately diluted cells in liquid YPD medium supplemented with 10% FBS, with shaking at 37 °C. Colony morphologies were assessed on YPD plates supplemented with 10% FBS. Approximately 20 cells of each strain were incubated on individual plates at 37 °C for 5–7 days before the photographic documentation of colonies.

### 2.4. Measurement of Cell Wall Composition

(1)Chitin quantification [28]

Cells collected in the exponential growth phase were treated with calcifluor white (CFW) (50 μg/mL) for 10 min. After 3 washes with phosphate-buffered saline, their fluorescence was evaluated using a Varioskan Flash spectral-scanning multimode reader (Thermo Fisher Scientific Inc., Waltham, MA, USA) at excitation and emission wavelengths of 325 and 435 nm, respectively. The presented data were standardized to cell concentrations assessed by optical density measurements.

(2)Determination of β-1,3-glucan concentrations [28,29]

Exponential phase cells were washed twice and resuspended in TE buffer to a final OD_600_ of 0.2–0.5. Subsequently, NaOH was added to a final concentration of 1.0 M, and the mixture was incubated at 80 °C for 30 min. Following the addition of 2.1 mL aniline blue solution (0.03% aniline blue, 0.18 M HCl and 0.49 M glycine/NaOH, pH9.5), the samples were incubated at 50 °C for 30 min, and then 30 min at room temperature for fluorochrome reaction and decolorization. Fluorescence was measured using the Varioskan Flash spectral-scanning multimode reader (Thermo Fisher Scientific Inc., Waltham, MA, USA) at the excitation and emission wavelengths of 400 and 460 nm, respectively.

(3)Alcian blue staining and determination of phosphomannan content [28,30]

Exponential phase cells were harvested via centrifugation and washed with 1 mL HCl (0.02 M). The cells were resuspended in 1 mL Alcian blue (50 μg/mL in 0.02 M HCl), incubated at room temperature for 10 min, and centrifuged. The OD_600_ of the supernatant was measured using a spectrophotometer (Unico Instrument Co., Ltd., Shanghai, China).

### 2.5. CFW Staining of Chitin and Fluorescence Microscopy

The staining of the cell wall chitin with CFW was conducted following established procedures [31]. Exponential phase cells were fixed with neutral-buffered formalin (10% (*v*/*v*), Sigma-Aldrich) and stained with CFW (25 μg/mL). 

### 2.6. Quantitative Reverse Transcription PCR Analysis of CaCHS1, CaCHS2, CaCHS3, and CaCHS8

Exponential phase cells were collected by centrifugation, washed twice with sterile water, and rapidly frozen in liquid nitrogen. Total RNA extraction was carried out using the MiniBEST Universal RNA extraction kit (TaKaRa, Dalian, China), following the manufacturer’s protocols, with DNaseI treatment to eliminate DNA contamination. The total RNA was utilized for cDNA synthesis using the PrimeScript RT master mix (TaKaRa). Quantitative PCR assays used TB-Green-Premix-Ex-Taq-II (TaKaRa). The expression levels of *CaCHS1*, *CaCHS2*, *CaCHS3*, and *CaCHS8* were assessed using specific primer pairs (Table 2), while the CaACT1 transcript served as an internal control [18]. The change in gene expression was evaluated using the 2^−△△CT^ approach [23].

### 2.7. Protein Extraction and Western Blotting

*Ca* protein extracts were prepared using a previously described protocol [32,33]. Lysates from OD_600_ = 1 of cells were subjected to SDS-PAGE and immunoblotting with the anti-phospho-p44/42MAPK (Thr^202^/Tyr^204^) antibody (Cell Signaling Technology (CST), Danvers, MA, USA) to concomitantly assess the phosphorylation levels of CaCek1 and CaMkc1. The probing of immunoblots using the anti-α-tubulin antibody (Novus Biologicals, Centennial, CO, USA) provided a loading control.

### 2.8. Virulence Assay

Virulence assays were conducted following established procedures [22,23]. Seven-week-old male BALB/c mice (weight = 20–22 g) were maintained in individually ventilated cages. The mice (n = 12/strain) were intravenously inoculated with 1 × 10^6^ cells/mL in 0.9% (*w*/*v*) NaCl solution/mouse via the lateral tail vein. Animals (n = 2/strain) were randomly euthanized following 48 h of injection, and the CFUs in their kidneys were quantified on SD-URA plates. Mouse survival rates were determined over 30 days, and fungal infiltration in the mouse kidneys was assessed through histopathological evaluation, as described previously [22,23]. The histological analysis of the collected mouse kidneys was performed, and stained sections were analyzed using a Axio Imager 2 (ZEISS Microscopy, Jena, Germany). The mouse experiments were conducted following the protocols approved by the Ethics Committee of Huaibei Normal University, China (permission ECHN191016).

### 2.9. Statistical Analysis

Statistical tests were conducted using SPSS v19.0. *p* < 0.05 was deemed significant.

## 3. Results

### 3.1. Genomic Data Analysis

The sequences of *Ca orf19.1034* came from the *Ca* genomic database (http://www.candidagenome.org (accessed on 21 March 2018)). *orf19.1034* spans 552 base pairs and codes for a protein consisting of 183 amino acids. According to SMART analysis (http://smart.embl-heidelberg.de (accessed on 21 March 2018)), the protein encoded by *orf19.1034* harbors one transmembrane region, commencing at position 13 and terminating at position 32 of the primary sequence, along with one cytochrome *b*5-like heme-binding domain, spanning from position 62 to 162 (Appendix A). The protein encoded by *orf19.1034* exhibited 27.6% and 31.6% sequence similarity to *S. cerevisiae* ScDap1 and *Candida glabrata* CgDap1, respectively. *ScDap1* and *CgDap1* also feature cytochrome *b*5-like heme-binding domains (Appendix A). Consequently, *orf19.1034* of *Ca* was designated as *CaDAP1*.

### 3.2. The Deletion of CaDAP1 Renders Ca Cells Susceptible to Caspofungin and Terbinafine

To assess the role of *CaDAP1* in *Ca*, we obtained homozygous deletion mutants of *CaDAP1* by sequentially replacing its two alleles with the *SAT1*-flipper and *HIS1* cassette (Figure 1). We tested the sensitivity of the *Cadap1/Cadap1* mutant to fluconazole, caspofungin, and terbinafine. Unlike the wild-type (WT) RM1000, the *Cadap1/Cadap1* mutant was sensitive to 0.25 μg/mL caspofungin and 5 μg/mL terbinafine (Figure 2), but remained insensitive to 5 μg/mL fluconazole (Appendix A). The reintroduction of *CaDAP1* into the homozygous mutant cells reversed these sensitive phenotypes. Thus, it is suggested that CaDap1 plays critical roles in conferring resistance to terbinafine and caspofungin in *Ca*.

### 3.3. The Deletion of CaDAP1 Reduces the Chitin Content in Cell Walls, and Downregulates the Phosphorylation Levels of CaMkc1

The echinocandin drug inhibits β-1,3-glucan synthase, a pivotal enzyme tasked with synthesizing β-1,3-D-glucan, a primary component of cell walls [34,35]. The sensitivity of the *Cadap1/Cadap1* mutant to caspofungin suggested the involvement of *CaDAP1* in the cell wall stress response of *Ca*. We examined the phenotype of *Ca* cells lacking *CaDAP1* following exposure to CR and CFW, as well as to SDS. Unlike the WT and re-integrant strain (RS), *Ca* cells lacking *CaDAP1* displayed significant resistance to CR and CFW but sensitivity to SDS (Figure 3A). To elucidate the impact on the cell wall, we assessed the levels of cell-wall components, such as phosphomannan, β-1,3-glucan, and chitin. Compared with the WT, the chitin content was reduced by 50% (Figure 3B), while the phosphomannan content increased by 9% in the *Cadap1/Cadap1* mutant (Figure 3C); the β-1,3-glucan content was unchanged (Appendix A). These findings suggest that the loss of *CaDAP1* alters the cell wall composition.

To further explore the impact of the *CaDAP1* knockout on chitin content in cell walls, *Ca* cells were stained with CFW to visualize chitin. The intensity of CFW fluorescence accurately reflects the relative chitin content [31,36]. Compared with the WT and RS, the intensity of CFW fluorescence was significantly decreased in the *Cadap1/Cadap1* mutant (Figure 3D). This provides additional evidence that the *CaDAP1* knockout reduces the chitin content in *Ca* cell walls. 

To investigate whether the reduction in chitin content in the *Cadap1/Cadap1* mutant corresponds with a decrease in the expression of chitin synthase genes, quantitative RT-qPCR was used to detect the expression of *CaCHS1*, *CaCHS2*, *CaCHS3*, and *CaCHS8*, responsible for synthesizing the chitin content of *Ca* cell walls [37]. Interestingly, *CaDAP1* deletion decreased *CaCHS3* expression by 65% compared to the WT and re-integrant strains (Figure 3E), while the expression of *CaCHS1*, *CaCHS2*, and *CaCHS8* remained unchanged. These findings collectively suggest that *CaDAP1* plays an essential role in maintaining chitin content within the cell wall of *Ca*. 

Previous research has indicated the role of the PKC-CaMkc1 axis in regulating chitin biosynthesis [36]. Therefore, we investigated the impact of *CaDAP1* deletion on CaMkc1-mediated CWI signaling. Compared to the WT and RS, the phosphorylation level of CaMkc1 was decreased in the *Cadap1/Cadap1* mutants (Figure 3F). Additionally, the phosphorylation level of CaCek1 increased slightly in the *Cadap1/Cadap1* mutants compared with the WT and RS (Figure 3F).

### 3.4. Impact of CaDAP1 Deletion on Hyphal Development and Colony Morphology

To explore the role of *CaDAP1* in hyphal formation, filamentous growth was assayed in YPD medium supplemented with 10% FBS. Following induction with 10% FBS, the mean filament length of WT RM1000 was 41.9 μm (3 h induction; n = 174) (Figure 4A,B). The *dap1/dap1* mutant exhibited an average filament length of only 28.3 μm (3 h induction; n = 163), whereas the revertant strain reached 41.7 μm (3 h induction; n = 159) (Figure 4A,B). These results suggest that *CaDAP1* deletion impairs FBS-induced filamentation in *Ca* cells, a defect that can be reversed by reintroducing the *CaDAP1* gene (Figure 4A,B). Moreover, on the solid YPD medium, the *Cadap1/Cadap1* mutant formed colonies with a smoother surface compared to the wrinkled-surface colonies of the WT RM1000. The reintroduction of *CaDAP1* into the homozygous mutant restored colony morphologies similar to the WT strain (Figure 4C). Thus, the deletion of *CaDAP1* significantly affected filamentation and colony morphology in *Ca*.

### 3.5. The Deletion of CaDAP1 Affects the Virulence of Ca

To assess the involvement of *CaDAP1* in the virulence of *Ca* cells, mice were intravenously injected with WT RM1000, *Cadap1/Cadap1* mutant, or RS cells. At day 9, no survivors were observed among the groups of 10 mice injected with the WT strain, whereas for the *Cadap1/Cadap1* mutant, no survivors were observed by day 14, and for the RS by day 10 (Figure 5A). The CFUs/g in wet kidney tissues of mice correlated with the virulent effects of *Ca* after 48 h of infection: 4.35 × 10^6^ for the WT RM1000 (n = 2), 1.21 × 10^5^ for the *Cadap1/Cadap1* mutant (n = 2), and 4.42 × 10^6^ for the RS (n = 2). The microscopic examination of kidney tissues revealed a similar pattern to those infected with the WT RM1000 and RS, with heavy infiltration by hyphal filaments. In contrast, those infected with the *Cadap1/Cadap1* mutant exhibited fewer filaments, albeit in a similar pattern, along with yeast-like fungal cells. Those injected with saline buffer (CK) exhibited no *Ca* cells (Figure 5B). Collectively, these findings suggest that *CaDAP1* deletion markedly impacts the virulence of *Ca* cells.

## 4. Discussion

In *S. cerevisiae*, ScDap1 is essential for ergosterol biosynthesis and confers resistance to azoles [14,15,16,17,18]. In the fungal pathogen *C. glabrata*, CgDap1 is involved in mediating azole tolerance [38]. In the human pathogen *Aspergillus fumigatus*, AfDapA is necessary for ergosterol biosynthesis and azole resistance [20]. The present investigation has identified and characterized the *CaDAP1* gene. Similar to the orthologs in *C. glabrata* and *S. cerevisiae*, CaDap1 contains a cytochrome *b*5-like heme-binding domain (Appendix A). Our experiments demonstrated that *CaDAP1* deletion renders *Ca* cells susceptible to terbinafine and resistant to fluconazole (Figure 2). This outcome implies that the response mechanism of Cadap1 to fluconazole stress is different from those of Scdap1, Cgdap1, and AfdapA. Additionally, we found that the deletion of *CaDAP1* increased the sensitivity of *Ca* to SDS-induced cell membrane stress (Figure 3A), indicating a potential alteration in the composition of the cell membrane due to *CaDAP1* deletion. 

Chitin is a critical component of fungal cell walls and septa, playing pivotal roles in maintaining fungal pathogenicity and adaption to stress [39,40,41]. A previous study showed that a *Ca* mutant with low chitin levels exhibited increased resistance to CFW and elevated sensitivity to SDS [42]. Our study reveals that the absence of *CaDAP1* also leads to cell resistance to CFW and sensitivity to SDS. In addition, the chitin content was reduced in the *Cadap1/Cadap1* mutant cell walls. In *Ca*, chitin synthesis is facilitated by four chitin synthase enzymes encoded by *CaCHS1*, *CaCHS2*, *CaCHS3*, and *CaCHS8*, with CaChs3 primarily responsible for chitin synthesis [40,43,44]. Compared to the WT strain, the expression level of *CaCHS3* was remarkably decreased in the *Cadap1/Cadap1* mutant. Munro et al. demonstrated an obvious reduction in chitin content in the *Camkc1/Camkc1* mutant, implying the post-transcriptional regulation of CaChs3 through PKC signaling [36,37]. Han et al. reported that the blocking of β-1,6-glucan synthesis triggers the phosphorylation of CaMkc1, leading to the activation of chitin synthase CaChs3 via a post-transcriptional mechanism, thereby maintaining cell wall chitin levels to ensure cell viability [39]. The knockout of *CaDAP1* decreases the CaMkc1 phosphorylation level, which may result in a reduced expression level of *CaCHS3* and/or the post-transcriptional regulation of CaChs3. Alternatively, *CaDAP1* may serve as a transcriptional activator of *CaCHS3* gene expression, and its deletion may affect the transcriptional activation of *CaCHS3,* leading to decreased chitin levels in the *Cadap1/Cadap1* mutant. Previous studies have indicated that elevating chitin content in the *Ca* cell wall can enhance cell resistance to echinocandins [31,45,46]. The *Cadap1/Cadap1* deletion strain is sensitive to caspofungin, probably due to reduced chitin content in the cell wall. 

Rowbottom and co-workers reported that *CaBNI4* mutants exhibit low cell-wall chitin levels and display smooth colony morphology on serum agar [42]. Compared to the wild-type strain, the *Cadap1/Cadap1* mutant displayed smoother colonies on a serum-containing solid medium, possibly due to a reduced chitin level in the cell walls. A previous study has highlighted the necessity of normal chitin content in cell walls for the virulence of *Ca* [40]. Decreased chitin levels in the *Cadap1/Cadap1* mutant may potentially decrease the virulence of *Ca*. In conclusion, our research indicates that CaDap1 is indispensable for hyphal development, antifungal drug resistance, the maintenance of chitin content in the cell wall, and virulence. Thus, this investigation offers novel insights into the functional role of *CaDAP1* in *Ca*.

## Figures and Tables

**Figure 1 jof-10-00316-f001:**
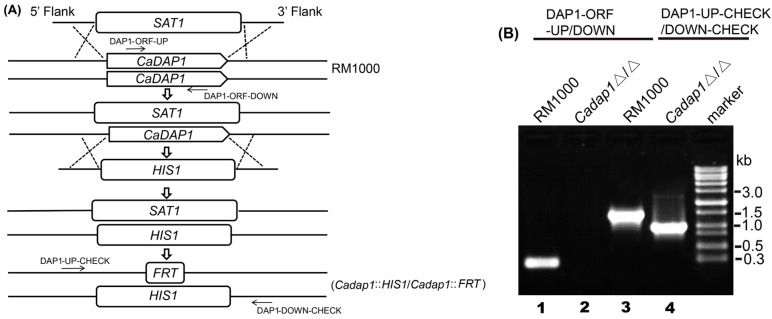
Deletion of two *CaDAP1* alleles. (**A**) Sequentially targeted disruption of two *CaDAP1* alleles in WT RM1000 to generate *Cadap1*Δ/Δ (*Cadap1*::*HIS1/Cadap1*::*FRT*). Strain designations are depicted on the right, while primer sites are highlighted with arrows. (**B**) PCR verification of the homozygous mutant *Cadap1*Δ/Δ genotypes. Lane 1: A 0.25 kb fragment from RM1000 was amplified using the primers DAP1 − ORF − UP/DOWN. Lane 2: No fragment was amplified from *Cadap1*Δ/Δ (*Cadap1*::*HIS1/Cadap1*::*FRT*) using the primers DAP1 − ORF − UP/DOWN. Lane 3: A 1.4 kb fragment was amplified from RM1000 using primers DAP1 − UP − CHECK/DOWN-CHECK. Lane 4: The 0.98 kb and 2.6 kb fragments containing *FRT* and *HIS1,* respectively, were amplified from *Cadap1*Δ/Δ using the primers DAP1 − UP − CHECK/DOWN − CHECK.

**Figure 2 jof-10-00316-f002:**
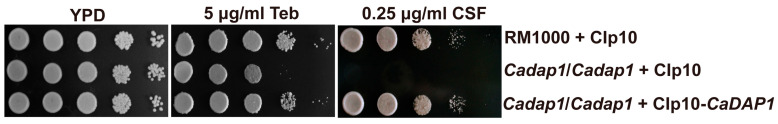
*CaDAP1* is involved in conferring resistance to terbinafine and caspofungin. Growth phenotypes of the WT, *Cadap1/Cadap1* mutant, and CaDAP1 RS were evaluated following exposure to caspofungin (0.25 μg/mL) and terbinafine (5 μg/mL). Overnight cell cultures were then grown in YPD to the exponential phase at 30 °C. Tenfold serial dilutions were spotted onto YPD plates supplemented with caspofungin or terbinafine, and incubated at 30 °C for 48–72 h.

**Figure 3 jof-10-00316-f003:**
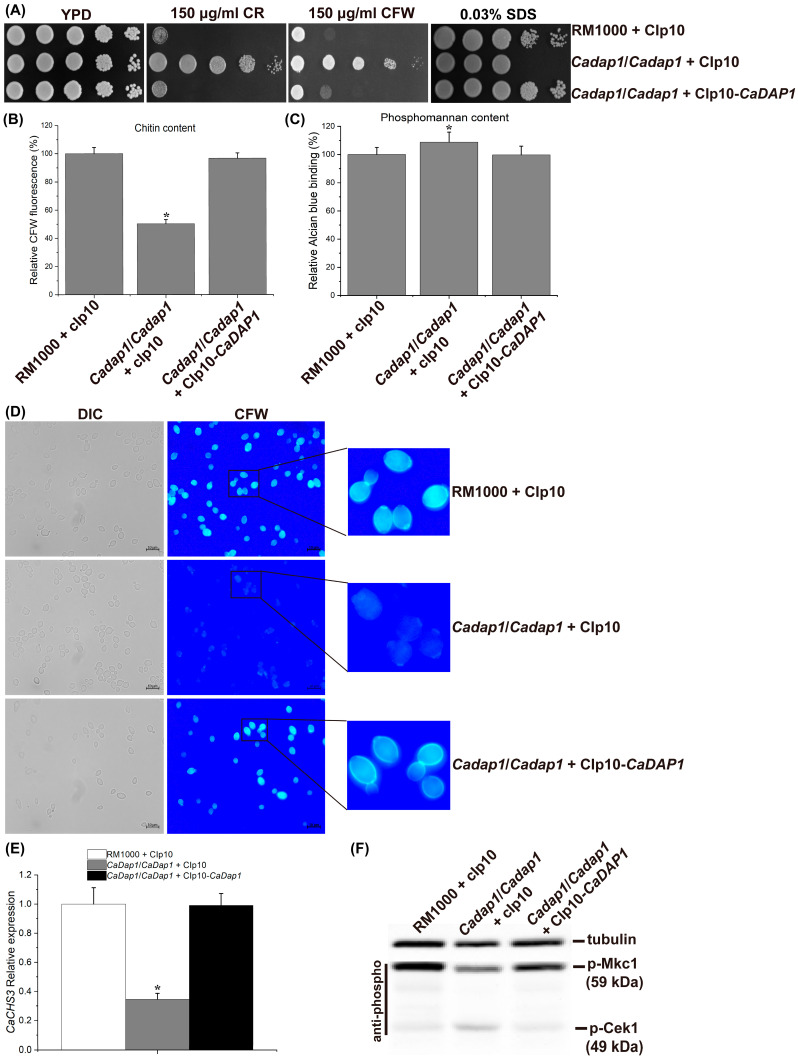
The deletion of *CaDAP1* leads to reduced chitin content in cell walls, accompanied by decreased phosphorylation levels of CaMkc1 and slightly increased phosphorylation levels of CaCek1. (**A**) The deletion of *CaDAP1* renders *Ca* cells resistant to CR and CFW, but sensitive to SDS. Growth phenotypes of WT, *Cadap1*/*Cadap1* mutant, and *CaDAP1* RS following exposure to 150 μg/mL CR and 150 μg/mL CFW, along with 0.03% SDS. Knockout of *CaDAP1* results in a 50% decrease in chitin content (**B**) and a 9% increase in phosphomannan content (**C**) of the cell wall. (**D**) The intensity of CFW fluorescence of WT, *Cadap1*/*Cadap1* mutant, and *CaDAP1* RS stained with 25 μg/mL CFW. (**E**) Loss of *CaDAP1* downregulates the expression level of *CaCHS3*. The WT strain (white bar), *Cadap1*/*Cadap1* mutant (gray bar), and RS (black bar) were cultured in YPD medium to the exponential phase at 30 °C. *CaCHS3* expression is depicted as a fold increase compared to the WT strain. (**F**) The deletion of *CaDAP1* downregulates the phosphorylation levels of CaMkc1 and slightly upregulates the phosphorylation levels of CaCek1. Exponential phase cultures were harvested, and protein extracts separated using SDS-PAGE, followed by immunoblotting with antibodies against phosphorylated CaMkc1 or CaCek1. Immunoblotting with antibodies against the α-tubulin served as a loading control. An asterisk (*) denotes a significant difference (*p* < 0.05) between the *Cadap1*/*Cadap1* mutant and WT or RS.

**Figure 4 jof-10-00316-f004:**
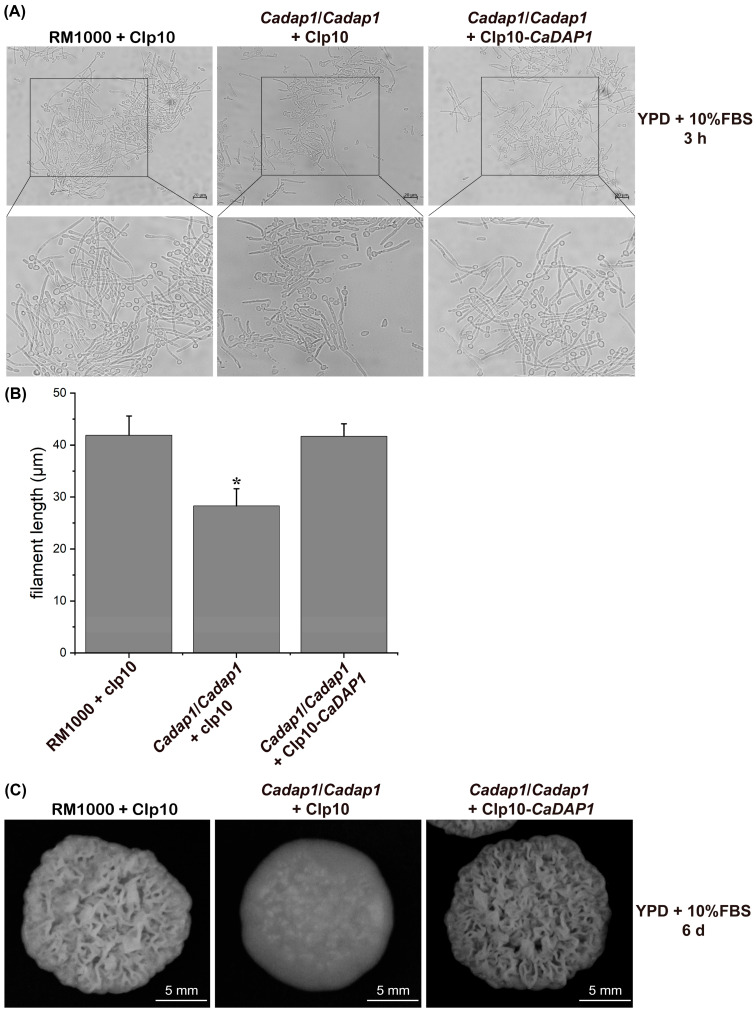
The deletion of *CaDAP1* affects hyphal development and colony morphology. Induction of filamentation was visualized in YPD medium supplemented with 10% FBS for both WT RM1000 and the *Cadap1/Cadap1* mutant, which contained either the integrated CIp10 vector or CIp10-*CaDAP1* plasmid (**A**). Filament length was quantified for each strain (**B**), and their colony-formation ability assessed (**C**). The difference in filament length between WT RM1000 and *Cadap1/Cadap1* mutant (**B**) was significant. An asterisk (*) denotes a significant difference (*p* < 0.05) between the Cadap1/Cadap1 mutant and WT or RS.

**Figure 5 jof-10-00316-f005:**
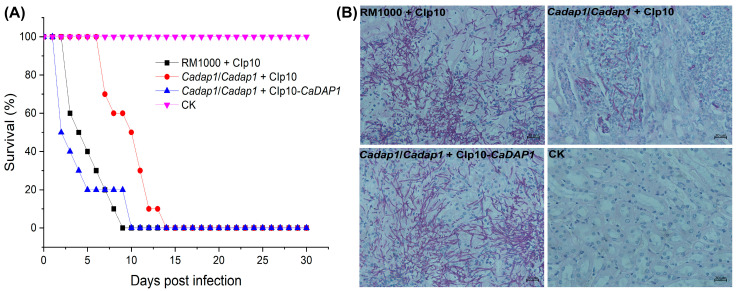
Assessment of virulence. (**A**) The survival rates of mice (n = 10) were evaluated after infection with WT RM1000, homozygous *Cadap1*/*Cadap1* mutant, and RS. Daily monitoring for morbidity and survival tracking was conducted over a 30-day period. (**B**) Kidney tissues from mice infected with WT strain, *Cadap1/Cadap1* homozygous mutant, and RS were subjected to histopathological examination. Saline buffer injection (CK) served as a negative control. Sections from infected kidney tissues were stained with periodic acid–Schiff’s reagent. Representative images of 5 kidney cross-sections (n = 2 mice/strain) were captured at 40× magnification.

**Table 1 jof-10-00316-t001:** Strains and plasmids used in this study.

Strains and Plasmids	Genotype	Source
*C. albicans* strains		
RM1000	*ura3*::*λimm434*/*ura3*::*λimm434 his1*:: *hisG/his1*::*hisG*	[24]
*Cadap1*/*Cadap1*	RM1000 *Cadap1*::*HIS1*/*Cadap1*:: *FRT*	This study
RM1000+CIp10	RM1000 *RPS1*/*rps1*::CIp10	[23]
*Cadap1*/*Cadap1*+CIp10	RM1000 *Cadap1*::*HIS1*/*Cadap1*:: *FRT RPS1*/*rps1*::CIp10	This study
*Cadap1*/*Cadap1*+ CIp10-*CaDAP1*	RM1000 *Cadap1*::*HIS1*/*Cadap1*:: *FRT RPS1*/*rps1*::CIp10-*CaDAP1*	This study
Plasmids		
pSFS2	*SAT1* flipper cassette, Amp^r^	[25]
pGEM-*HIS1*	*HIS1* cassettes, Amp^r^	[26]
CIp10	Integrating vector for *C*. *albicans*, *URA3*, Amp^r^	[27]
CIp10-*CaDAP1*	Full-length *CaDAP1* gene in CIp10	This study

**Table 2 jof-10-00316-t002:** Primers used in this study.

Primer Name	Sequence (5’–3’)	Restriction Site
DAP1-NAT-UP	TGGAGACTACATATATTAAGTCACATATATAAATGCACGTACTTTTTTTTTCTTCACACACACCACCAAACATACTAACGCGATCCTTAAATCAACCATCAATTAACCCTCACTAAAGGG	
DAP1-NAT-DOWN	AATATAGACCTAGATATATGCATAACGTTTTTTAATTTTTCCCTATTAATATTATTAAATGTCTCTATTAATGCATTCCGGGGAACTTCATATGTTCACAAATACGACTCACTATAGGG	
DAP1-HIS-UP	TGGAGACTACATATATTAAGTCACATATATAAATGCACGTACTTTTTTTTTCTTCACACACACCACCAAACATACTAACGCGATCCTTAAATCAACCATCGGCCAGTGAATTGTAATACG	
DAP1-HIS-DOWN	AATATAGACCTAGATATATGCATAACGTTTTTTAATTTTTCCCTATTAATATTATTAAATGTCTCTATTAATGCATTCCGGGGAACTTCATATGTTCACACAAGTTGAACTCCCTTATGG	
DAP1-UP-CHECK	CAAGAGTCAAGAAAGGAAAGC	
DAP1-DOWN-CHECK	TGAAGGTGGTGGTTATGGAG	
DAP1-ORF-UP	TTCAACATGGGGGTATCGAG	
DAP1-ORF-DOWN	CCAGTAACATAAACTCTAGCTGC	
DAP1-UP-CLO	CGCGAGCTCTACCACCCCTTCCTTACATC	*Sac*I
DAP1-DOWN-CLO	AAGGAAAAAAGCGGCCGCAACGCAGTAAATCTTCACAAGG	*Not*I
qCaACT1-F	GAAGCCCAATCCAAAAGAG	
qCaACT1-R	CTTCTGGAGCAACTCTCAATTC	
qCaCHS1-F	CTCTCGAGAAACATTTGCTG	
qCaCHS1-R	GTAGTTTCAGGACTGGCATC	
qCaCHS2-F	CACTTCTCAAACACAGATCC	
qCaCHS2-R	CATGAGATGATTAGGTTGACC	
qCaCHS3-F	TCACCCAGATGTTGTTCCTC	
qCaCHS3-R	GAGAATTAGGGTAATCGGTGG	
qCaCHS8-F	TGTTAGAAGCTGGTGGAGTC	
qCaCHS8-R	CATTTAAGTGGACGGAAACTC	

## Data Availability

The data that support the findings of this study are available from the corresponding author (xudayonghello@163.com or rx2500@163.com) upon reasonable request.

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
