# Peer review of "The Putative Cytochrome b5 Domain-Containing Protein CaDap1 Homologue Is Involved in Antifungal Drug Tolerance, Cell Wall Chitin Maintenance, and Virulence in Candida albicans"

_jof, 2024, doi:10.3390/jof10050316_

Round 1
Reviewer 1 Report
The authors demonstrate for the first time that in Candida albicans deletion of the putative cytochrome b5 domain-containing protein Dap1 leads to increased susceptibility to antifungals (caspofungin and terbinafine). In addition deletion of DAP1 leads to a decrease in chitin content and increased resistance to the cell wall stresses CFW and congo red. The alterations in the cell wall of the DAP1 deletion strain also result in a decrease activation of the cell wall salvage pathway. Inhibition of DAP1 was also shown to result in a decrease in virulence as a result of a defect in hyphal formation. There are some minor issues within the manuscript that need altered but overall this is a robust study of the effect of deleting DAP1 on C. albicans.
The Introduction is quite short, some sections are mentioned i the results of the paper which should be in the Introduction.
Line 90-91: The concentration that the cells were diluted to should be included.
Lines 96 - 116 (Section 2.4 in Methods): In the CFW staining section cells are described as being in exponential phase but for b-glucan and alcian blue staining logarithmic phase is used. Please can they be changed so that the same term are used. This occurs throughout the paper, please can it be amended so it's consistent.
Section 2.9 Statistical analysis - The authors should include details of exactly which statistics were actually used.
Line 173 - The word 'could' should be removed form the title.
Legend for Figure 2 (lines 197-202) does not mention
Lines 207-209: This part should have been mentioned in the Introduction.
In section 3.3 it might be interesting to present the b-glucan data so that all of the cell wall analysis is presented, or at least as supplementary data.
Figure 3D: The quality of the fluorescent images with the CFW staining to show changes in chitin are quite poor because there is a lot of autofluorescence. Are the authors able to improve the images?
Line 267: " a less rough surface" should be changed to "a smoother surface"
In Figure 4A can the authors improve the images so that it is easier to see the differences in the hyphae?
For Figure 4, the figure legend should include the size of the scale bar for the images in Fig. 4A.
Figure 4C should have scale bars.
In the legend for Figure 5 the authors should include what day the images in Fig. 5B were taken at.
It is a bit unclear where the authors found that the deletion of CaDAP1 leads to resistance to fluconazole or no change in susceptibility. Both options are implied in the text. It may be beneficial for the authors to include the fluconazole data in Fig. 2 to avoid confusion.
Author Response
|
Comments 1: The introduction is quite short, some sections are mentioned in the results of the paper which should be in the Introduction. Response 1: Revision: The cell wall of Ca is located in the outermost layer and is composed of chitin, β-glucan, mannan and cell wall proteins [7]. Chitin and β-glucan are located in the interior of the cell wall and form the core skeleton structure. The outermost cell wall is mainly composed of mannan, which masks the internal β-glucan and reduces the recognition of Ca by the host immune system. The cell wall of Ca plays a key role in maintaining cell integrity, morphogenesis, responding to changes in environmental conditions, interaction with the host, and pathogenesis [7,8]. Moreover, the cell wall of Ca is a prime target for antifungal drugs such as Echinocandins [9]. Despite its tough cell wall, Ca can flexibly change the relative levels of chitin, β-glucan, and mannan in response to environmental changes [10,11]. Thus, this potential remodeling of cell wall components is critical for maintaining Ca cell wall integrity (CWI) and is regulated by multiple signaling pathways, including the Mkc1, Hog1, and Cek1 mitogen-activated protein (MAP) kinase cascade [12]. Chitin, a component of the cell wall of Ca, is absent in humans and other vertebrates. Therefore, chitin synthesis is an excellent potential target for the development of antifungal drugs [13]. Ca regulates the expression of chitin synthase and the content of chitin in cell wall, through HOG, PKC and the Ca2+/calcineurin signalling pathways in response to environmental stress [13]. Thank you for pointing this out. We agree with this comment. Therefore, we have added the research on the cell wall of C. albicans in the introduction. The added content is in Lines 35-51 of the revised manuscript.
|
|
Comments 2: Line 90-91: The concentration that the cells were diluted to should be included. |
|
Response 2: Revision: the cell concentrations were of 5×107, 5×106, 5×105, 5×104, and 5×103 cells/ml. Agree. In the revised manuscript, we have added content about the cell concentration of the C. albicans diluent. This change can be found in lines 104-105.
|
|
Comments 3: Lines 96 - 116 (Section 2.4 in Methods): In the CFW staining section cells are described as being in exponential phase but for b-glucan and alcian blue staining logarithmic phase is used. Please can they be changed so that the same term are used. This occurs throughout the paper, please can it be amended so it's consistent. |
|
Response 3: Agree. We have used “exponential phase” uniformly in the revised manuscript. This change can be found in lines 123, 131, 138, 218.
Comments 4: Section 2.9 Statistical analysis - The authors should include details of exactly which statistics were actually used. |
|
Response 4: In this manuscript, the cell wall components mannan, β-1,3-glucan and chitin, as well as CaCHS3 expression and filament length, were all statistically analyzed by SPSS software.
Comments 5: Line 173 - The word 'could' should be removed from the title. |
|
Response 5: Agree. In the revised manuscript, the word 'could' has been removed. This change can be found in line 191.
Comments 6: Legend for Figure 2 (lines 197-202) does not mention Response 6: Sorry, we didn't get the meaning.
Comments 7: Lines 207-209: This part should have been mentioned in the Introduction. Response 7: Agree. In the introduction of the revised manuscript, we have added the content of cell wall components and antifungal drug echinocandin. This change can be found in lines 35-51 in the revised manuscript.
Comments 8: In section 3.3 it might be interesting to present the β-glucan data so that all of the cell wall analysis is presented, or at least as supplementary data. Response 8: Agree. The β-glucan data (Supplemental figure 3) has been presented in the supplemental file.
Supplemental figure 3. Deletion of CaDAP1 leads to unchanged β-1,3-glucan content in cell walls.
|
|
Comments 9: Figure 3D: The quality of the fluorescent images with the CFW staining to show changes in chitin are quite poor because there is a lot of autofluorescence. Are the authors able to improve the images? Response 9: Agree. We have improved the quality of the fluorescent images with the CFW staining. This change can be found in lines 258-259 in the revised manuscript.
Figure 3D. The intensity of CFW fluorescence of WT, Cadap1/Cadap1 mutant, and CaDAP1 RS after staining with 25 μg/ml CFW.
|
|
Comments 10: Line 267: " a less rough surface" should be changed to "a smoother surface". Response 10: Agree. " a less rough surface" has been changed to "a smoother surface". This change can be found in line 286 in the revised manuscript.
|
|
Comments 11: In Figure 4A can the authors improve the images so that it is easier to see the differences in the hyphae? Response 11: Agree. Figure 4 has been enlarged so that the difference in hyphae can be clearly observed. This change can be found in lines 290-291 in the revised manuscript.
Figure 4 A. Deletion of CaDAP1 affects hyphal development on YPD + 10% FBS medium. Induction of filamentation was conducted for both WT RM1000 and the Cadap1/Cadap1 mutant, which contained either the integrated CIp10 vector or CIp10-CaDAP1 plasmid, in YPD medium supplemented with 10% FBS.
Comments 12: For Figure 4, the figure legend should include the size of the scale bar for the images in Fig. 4A. Response 12: The figure legend has been included the size of the scale bar for the images in Figure 4A.
Comments 13: Figure 4C should have scale bars. Response 13: Agree. Figure 4C has been added with scale bars. This change can be found in lines 290-291 in the revised manuscript.
Figure 4C. Deletion of CaDAP1 affects colony morphology on YPD + 10% FBS medium.
Comments 14: In the legend for Figure 5 the authors should include what day the images in Fig. 5B were taken at. Response 14: Virulence assays were conducted over 30 days. For WT RM1000, homozygous Cadap1/Cadap1 mutant, and its complemented strain, histological analysis of the collected kidneys of dying mouse was performed, and stained sections were analyzed under a ZEISS microscope. The 0.9% (w/v) NaCl solution was used as control, and the kidney of mice on the 30th day was collected for analysis. Comments 15: It is a bit unclear where the authors found that the deletion of CaDAP1 leads to resistance to fluconazole or no change in susceptibility. Both options are implied in the text. It may be beneficial for the authors to include the fluconazole data in Fig. 2 to avoid confusion. Response 15: Agree. The fluconazole phenotype (Supplemental figure 2) has been presented in the supplemental file.
Supplemental figure 2. The Cadap1/Cadap1 mutant exhibited insensitive to fluconazole (Flu). Growth phenotypes of the WT, Cadap1/Cadap1 mutant, and CaDAP1 RS were evaluated following exposure to 5 μg/ml fluconazole. The cells cultured overnight were renewed in YPD medium and grown to exponential phase at 30°C. Following this, they underwent a tenfold serial dilution and were then spotted sequentially onto plates supplemented with fluconazole. Phenotypic characteristics were examined following incubation at 30°C for 48-72 hours.
|
Reviewer 2 Report
The MS by Xu and co-workers describe the identification and characterization of the function of a Candida albicans DAP1 (CaDAP1). The MS is well written although I advise a careful reading to correct some misspelling. The work was elegantly planned to answer the question of the authors. Not only they compared the gene with homologs of other species, constructed a mutant in which proved the loss of function, recovered once the gene is reintegrated. The results clearly show that this gene confers resistance to caspofungin and to terbinafine and that the deletion mutant suffers from a significant decrease in chitin. Moreover, not only they demonstrate that the loss of CaDAP1 leads to shorter hyphae, but also, using an in vivo assay, prove that the deleted mutant is less virulent.
Other comments/suggestions:
Abstract and throughout the document:
Why using the abbreviation “CA”? it sounds strange… the species name abbreviated is C. albicans and in fact there are authors that use the “Ca” (“C” from Candida and “a” from albicans) whenever they have to shorten a text. I believe this is not the case and if using an abbreviation you should use “Ca”. In fact the authors use Ca in the designation of the proteins and genes.
Line 88: the growth with antifungals is made with solid media? The authors did not use liquid media?
Line 105: “twice” instead of “thrice”
Line 109 and throughout the text: room temperature instead of ambient temperature
Fig. 4A – these photos should be improved; we hardly can see the fungal structures
Author Response
|
Comments 1: Abstract and throughout the document: Why using the abbreviation”CA”? it sounds strange… the species name abbreviated is C. albicans and in fact there are authors that use the “Ca” (“C” from Candida and “a” from albicans) whenever they have to shorten a text. I believe this is not the case and if using an abbreviation you should use “Ca”. In fact the authors use Ca in the designation of the proteins and genes. Response 1: Thank you for pointing this out. We agree with this comment. Therefore, we have changed "CA" with" Ca" in the revised manuscript.
|
|
Comments 2: Line 88: the growth with antifungals is made with solid media? The authors did not use liquid media? |
|
Response 2: Yes. In this study, solid medium was used to determine the growth phenotype of the strains against antifungal drugs, and no liquid medium was used to determine the growth phenotype. The solid media phenotype was able to demonstrate the sensitivity of the CaDAP1 deletion strain to the drugs.
|
|
Comments 3: Line 105: “twice” instead of “thrice”. |
|
Response 3: We have changed “thrice” with “twice”. This change can be found in line 123 in the revised manuscript.
Comments 4: Line 109 and throughout the text: room temperature instead of ambient temperature. |
|
Response 4: We have changed “ambient temperature” with “room temperature”. This change can be found in lines 127, 132 in the revised manuscript.
Comments 5: Fig. 4A – these photos should be improved; we hardly can see the fungal structures |
|
Response 5: Agree. Figure 4 has been enlarged so that the difference in hyphae can be clearly observed. This change can be found in lines 290-291 in the revised manuscript.
Figure 4 A. Deletion of CaDAP1 affects hyphal development on YPD + 10% FBS medium. Induction of filamentation was conducted for both WT RM1000 and the Cadap1/Cadap1 mutant, which contained either the integrated CIp10 vector or CIp10-CaDAP1 plasmid, in YPD medium supplemented with 10% FBS.
|